# Diabetic Muscle Infarction—A Rare Diabetic Complication: Literature Review and Case Report

**DOI:** 10.3390/ijerph20043390

**Published:** 2023-02-15

**Authors:** Maciej Rabczyński, Monika Fenc, Edwin Kuźnik, Paweł Lubieniecki, Agnieszka Hałoń

**Affiliations:** 1Department of Angiology, Hypertension and Diabetology, Wroclaw Medical University, Borowska 213, 50-556 Wrocław, Poland; 2Department of Radiology, Wroclaw Medical University, Borowska 213, 50-556 Wrocław, Poland; 3Department of Clinical and Experimental Pathology, Wroclaw Medical University, Marcinkowskiego 1, 50-368 Wrocław, Poland

**Keywords:** diabetic muscle infarction, insulin-dependent diabetes mellitus complication, diabetic microangiopathy

## Abstract

We present a case of a 31-year-old patient with type 1 diabetes diagnosed at the age of 6. Diabetes is complicated with neuropathy, retinopathy, and nephropathy. He has been admitted to the diabetes ward due to inadequate diabetes control. Gastroscopy and abdominal CT were performed, and gastroparesis was confirmed as an explanation for postprandial hypoglycemia. During hospitalization, the patient reported sudden pain localized on the lateral, distal part of his right thigh. The pain occurred at rest and was aggravated by movement. Diabetic muscle infarction (DMI) is a rare complication of long-lasting, uncontrolled diabetes mellitus. It usually occurs spontaneously, without any previous infection or trauma, and is often misdiagnosed clinically as an abscess, neoplasm, or myositis. DMI patients suffer from pain and swelling of the affected muscles. Radiological examinations, including MRI, CT, and USG, are most important for the diagnosis, assessing the extent of involvement and differentiating DMI from other conditions. However, sometimes a biopsy and histopathological examination are necessary. The optimal treatment has still not been determined. There is also a potential risk of DMI recurrence.

## 1. Introduction

Diabetes mellitus is a very common disease all over the world. The most common diabetic complications include nephropathy, neuropathy, and macro- or microangiopathy. Diabetic muscle infarction (DMI) is a rare complication of long-lasting, uncontrolled diabetes and was first described in 1965 [1]. The pathogenesis of diabetic muscle infarction is not fully known. Possible causes include, in addition to poorly compensated diabetes and its complication of vascular microangiopathy, arteriosclerosis with embolic material from ulcerated plaques [2]. Silberstein et al. proposed a mechanism based on hypoxia and reperfusion [3].

It usually occurs spontaneously, without previous infection or trauma, and is often misdiagnosed clinically as an abscess, neoplasm, or myositis. DMI patients suffer from pain and swelling of the affected muscles. DMI is not included in most of the standard text publications. Due to the small number of cases, it is difficult to determine the natural history and the most appropriate method of diagnosis and treatment. In this paper article, we present a case of DMI in a dialysis patient with type 1 diabetes mellitus.

## 2. Case Report

We present a case of a 31-year-old patient with type 1 diabetes diagnosed at the age of 6. Diabetes is complicated by neuropathy; a symmetrical distal sensory polyneuropathy with loss of sensation of touch, vibration, temperature, and pain was diagnosed three years ago. Its complication is Charcot osteoarthropathy, for which reason the patient is advised to relieve pressure on his lower limb and uses a wheelchair outside the home. At home, the patient moves around independently, but due to his advanced retinopathy, he often suffers small bruises, especially on his lower extremities.

The patient was also diagnosed with proliferative diabetic retinopathy (PDR) in both eyes. Due to the rapid progression of this microvascular complication, the patient lost his vision in the left eye. His poor visual acuity was also considered to be the reason for mobility problems, an increased number of injuries, and difficulties with applying dressings on diabetic foot ulcers. End-stage renal disease was diagnosed as a complication of nephropathy. The patient has been on hemodialysis for 3 years and during this time secondary hyperparathyroidism was diagnosed.

Currently, the patient has been admitted to the diabetes ward due to inadequate diabetes control. He has developed postprandial hyperglycemia but also hypoglycemia that occurred after meals and varied from 1.3 to 26.7 mmol/L. His glycosylated hemoglobin was 8.3% which was not reliable because of anemia, and recurrent symptomatic severe hypoglycemic episodes provoked by even low doses of insulin. The patient reported frequent use of glucagon to avoid hypoglycemia.

On admission to the hospital, the patient was in good general condition, with good cardiovascular and respiratory capacity, and on physical examination, there was a deformity of the left foot with a small (20 × 10 mm) ulceration present on the sole of the foot in the midfoot area, with no signs of infection—a picture of Charcot osteoarthropathy. In addition, there was discrete edema of both calves which we associate with dialysis therapy and a sedentary position. Small abrasions of the epidermis without signs of local inflammation were present on the skin of the lower extremities from small injuries sustained at home. There were no pressure sores or other skin ulcers. The muscle strength of the upper and lower limbs was normal and symmetrical. A pulse was present in typical areas. His heart rate was steady on ECG normal recording. Above the lungs without stasis, his abdomen was soft at chest level. His calculated BMI was 25.0. His blood pressure was not well controlled, and was measured on admission to be 180/100 mmHg. His laboratory results are presented in Table 1. The patient also complained of abdominal pain localized in the epigastric area. Gastroscopy and abdominal computed tomography (CT) were performed, and gastroparesis was confirmed as an explanation for postprandial hypoglycemia. In the treatment of diabetes, the patient received insulin in an intensive insulin therapy regimen with basal insulin administered once a day and injections of prandial insulins. We introduced a division of prandial insulin, i.e., administration of half the dose before a meal and after control in the second hour after the meal, a decision was made to administer the second part of the calculated dose. The overall intervention resulted in a significant improvement in postprandial glycaemic control.

Serum albumin concentration was 3.5 g/dL. In addition, low LDL cholesterol had been observed for a long time without the use of statins. BMI was normal and no features of cachexia were observed. Low LDL-cholesterol and albumin concentrations may have been provoked by malnutrition, which may have been influenced by the patient’s poor economic status, as well as disorders affecting the ability to absorb nutrients. In our patient, we identified the following conditions that could lead to malabsorption: gastroparesis and anemia.

During the next days of hospitalization, the patient reported sudden pain localized on the lateral, distal part of his right thigh. The pain occurred at rest and was aggravated by movement. The patient denied any prior trauma such as a blow, pressure, or the stretch of this leg region.

In a physical examination, we found a painful and swollen area on palpation. There was no erythema, warmth, or any other changes on the overlying skin. There were no signs of critical limb ischemia, and the pulse was palpable on both lower extremities. Doppler ultrasound (USG) was performed, which excluded vein thrombosis or leg ischemia. USG revealed a hypoechogenic oval structure sized 4.87 × 2.74 cm (Figure 1). We observed a small blood vessel running through the lesion. The image was not characteristic of fluid collection. Increasing inflammatory markers were linked to the presence of permcath. We took blood samples for bacterial culture. Biotraxone was administered intravenously. Due to intense leg pain, the patient required analgesics.

Initially, in the differential diagnosis, we considered focal myositis, abscess, or diabetic muscle infarction; therefore, magnetic resonance imaging (MRI) of the right leg with gadolinium contrast was considered. However, because of end-stage renal disease (ESRD), the nephrologist suggested a computed tomography instead. CT scans revealed an irregular area, not enhanced after contrast administration, located in the short head of the biceps femoris muscle with the size of 2.3 × 2.2 × 8.0 cm accompanied by local subcutaneous edema (Figure 2). The image raised suspicion of abscess or local necrosis. Moreover, lymphadenopathy in the right groin and along external iliac vessels was observed, as well as intramural calcification of arterial walls.

After a few days of antibiotic administration, due to no improvements in clinical symptoms and laboratory tests, we changed the treatment to clindamycin 3 × 600 mg. Nonetheless, the pain and restricted range of motion were still unchanged. The results of the blood culture were negative.

The decision of muscle biopsy for histopathological examination was made. The biopsy demonstrated diffuse areas of muscle necrosis and edema with a focal phenomenon of extravasation of blood. Light microscopy showed phagocytosis of necrotic muscle fibers with dispersed mononuclear cell infiltration. The presence of areas of granular tissue, early loose fibrotic tissue, and focal muscle fiber regeneration with lymphocytic infiltration was observed (Figure 3). Treatment involving bed rest and acetylsalicylic acid administration was introduced with subsequent improvement. The patient remained asymptomatic and was followed up as an outpatient.

After the USG doppler, we did not confirm vascular disorders such as deep vein thrombosis, leg ischemia, post-traumatic hemorrhage or aneurysms (the patient denied prior trauma). The patient was not treated with statins. According to the normal value of WBC, lack of fever, resting pain, local signs of infection such as skin warmness and redness, a bacterial infection was not proven. Antibiotic treatment was not effective. Sudden severe symptom onset was not typical for neoplasia.

It was very likely that the patient had suffered an unnoticed injury in this area. Hence, an extensive imaging diagnosis was carried out (USG, CT scan), and the procedures performed did not give a clear diagnosis.

To exclude other inflammatory problems, such as myositis or rhabdomyonecrosis, or amyotrophy, we decided to take a histopathological biopsy, which demonstrated diffuse muscle necrosis, edema, mononuclear cell infiltration, and local muscle fiber regeneration. The diagnosis of DMI was made after biopsy and histopathological evaluation of samples.

## 3. Discussion

Diabetic myonecrosis, otherwise termed diabetic muscle infarction (DMI), ischemic myonecrosis, and tumoriform focal muscular degeneration are rare complications of diabetes mellitus that occur often among adults with long-lasting, uncontrolled DM; however, some cases in children have been reported as well. The first cases of DMI were reported in 1965 by Angervall and Stener, who described two diabetic patients with initial suspicion of a tumor. Since then, about 228 cases have been reported in Embase.

According to a systematic review, 126 initial cases were described in the literature. In total, 54% of them were concerning females, which suggests a lack of or poor gender relationship. The mean age of all patients was 44.6 years. However, the average varied when it came to the type of DM. The mean age of patients suffering from T1DM was 35.9 years, while the mean age of those with T2DM was 52.2 years. DMI more often affected patients with T2DM (50%) than those with T1DM (41.7%). The mean duration of DM at the moment of diagnosis was visibly longer in T1DM patients and lasted 18.9 years (5–33), in comparison to 11 years in T2DM patients (1–25) [4].

There is a regularity in the clinical presentation of patients who developed DMI that corresponds strictly with the profile of our patient. Poor glycemic control is the most important factor in developing long-term diabetes-related complications such as nephropathy, peripheral neuropathy, and retinopathy. Especially frequent is nephropathy, which accompanied DMI in 75% of cases [4], including end-stage renal disease requiring hemodialysis. An episode of DMI should be taken into consideration in cases of any diabetic patient who presents with a sudden onset of acute muscular pain and palpable mass, firm on touch, tender, and located in an extremity. The muscle distortion most commonly affects the lower limb—front thigh (over 55% of cases; the quadriceps, adductors, hamstrings), then the calf (15%) and back thigh [4,5]. It can be uni- or bilateral and involve a single muscle or the whole muscle compartment. The least affected regions are the arm and forearm, but the reason for this predilection remains unclear [6]. The affected extremity presents no sign of infection, skin above the mass seems intact or presents delicate erythema. There is no trauma history or fever [7]. The pain appears at rest and increases with movement. Functional impairment is related to damaged muscle and local swelling.

In our review of the literature, we found no information on spontaneous smooth muscle or myocardial necrosis in diabetes mellitus. There are reports that skeletal muscle infarction in a diabetic population has a similar prognosis as compared with myocardial infarction. DMI is a very late-stage, terminal diabetic complication [8].

Standard laboratory examinations mostly do not reveal any specific pattern. Creatine kinase level can be elevated or remain normal as we observed in our patient. Classic inflammatory markers, such as CRP or ESR, have been reported to be elevated more often than WBC values, which remained within normal limits in most cases [4,6].

Besides clinical presentation, performing radiological imaging seems crucial to make a final diagnosis. The method of choice remains as MRI, especially with the administration of i.v gadolinium, even if the findings are not pathognomonic for DMI. Axial short tau inversion recovery (STIR) and fat-suppressed T2-weighted images show the increased signal intensity of affected muscle and ring-enhancing patterns around the area of necrosis after administration of contrast material. Sometimes, instead of this pattern, only minimal to modest enhancement was reported. Other characteristic features on MRI scans include an isointense to hypointense signal in T1-weighted images with associated subfascial fluid and subcutaneous or interfascial oedema. Hyperintensity on T1-weighted images more likely supports hemorrhagic infarction [7,9,10]. In our case, using gadolinium contrast was problematic because of coexisting nephropathy and potential risk of nephrogenic systemic fibrosis; so, in consequence, we decided not to perform the MRI. Since MR findings may pose a problem in distinguishing DMI from intramuscular abscess, inflammatory or autoimmune myopathies, in some cases, biopsy may be necessary.

Ultrasonography has also been recommended as an imaging diagnostic technique due to its common availability in an emergency setting. It allows for the rapid exclusion of DVT, abscess or necrotic tumor (an absence of internal motion or swirling of fluid transducer pressure; a lack of a predominantly anechoic area). Sonographic findings in DMI include a well-marginated, hypoechoic, intramuscular lesion. In addition, it can be followed by internal linear echogenic structures coursing through the lesion [7,11].

Computed tomography examination is less helpful. In our process of diagnostics, it left us with more questions than answers, but usually it may serve to exclude local abscesses, tumors, or bone destruction. The axial unenhanced CT image shows diffuse muscular enlargement with low attenuation, increased attenuation of the subcutaneous fat, and thickening of subcutaneous fascial planes and of the skin. After contrast administration, CT reveals a low-attenuation lesion with ring-enhancing margins in the involved muscle [7,11,12].

Muscle biopsy can assure a clear, conclusive diagnosis; nonetheless, it is not highly recommended because of the risk of potential complications related to the procedure, such as bleeding into the lesion, an extended recovery period, and infection. It should be reserved for atypical clinical presentations when the standard investigation methods require further confirmation. Even though the needle/incision biopsy has been advocated instead of the excision biopsy, it is not as well tolerated by the patient [13].

Vascular imaging studies of our patient do not confirm vascular disorders. He does not receive any statins.

Ineffective antibiotic therapy with negative infection parameters precludes bacterial infection. Sudden severe symptom onset was not typical for neoplasia.

Due to the inability to perform MRI, we decided to perform a biopsy which demonstrated diffuse muscle necrosis, edema, mononuclear cell infiltration, and local muscle fiber regeneration. The presence of focal blood extravasation does not differentiate hemorrhagic infarction from microvascular inflammation. It could be an ischemia–reperfusion injury.

It is controversial whether diabetic myonecrosis is in fact rare, or just seriously underrecognized due to its similarities with common musculoskeletal disorders and an absence of pathognomonic clinical and laboratory features. Although DMI itself is a relatively benign entity, it requires a judicious and vigilant differential diagnosis to eliminate the suspicion of another, sometimes more serious, condition such as DVT, abscess, cellulitis, osteomyelitis, benign and malignant tumors, pyomyositis, dermatomyositis, hematoma, focal myositis, ruptured Baker’s cyst, exertional muscle rupture, diabetic lumbosacral plexopathy, or diabetic amyotrophy. It is important, among other things, to rule out necrotizing fasciitis—an entity that occurs more often in individuals with DM and in which rapid surgical intervention is bound up with vital prognostics [14].

The doubts around the pathophysiology of diabetic myonecrosis led to several theories that have been put forth; however, none of them predominate. Chester and Banker suggested arteriosclerosis obliterans as the primary factor causing DMI. The authors further theorized that constriction of an infarcted, swollen area might account for the progressive nature of the lesion, sometimes involving the entire compartment [15]. Other authors consider the nature of DMI secondary to vasculitis, atherosclerosis or diffuse microangiopathy associated with hypoxia–reperfusion injury, a sequence of events that begins with a small thromboembolic material or intramuscular insulin injection. This leads to microvascular endothelial damage, and tissue ischemia which triggers an inflammatory cascade and reperfusion with extended production of reactive oxygen species causing more tissue damage and ischemic necrosis [3].

Some authors tend to link the known alterations of the coagulation–fibrinolysis system present in patients to DM that encompass increased levels of activated factor VII and fibrinogen, decreased levels of prostacyclin and tissue plasminogen activator, with a potential net effect of thrombogenesis. Moreover, Palmer and Greco described two cases of diabetic patients with end-organ microvascular complications who showed additional evidence of coagulopathy in the setting of detectable antiphospholipid antibodies. There is evidence based on epidemiological and genome-wide association studies that supports the correlation between T1DM and increased risk of the presence of anticardiolipin antibodies. Additional studies would reinforce or challenge the dependence of DMI on the prevalence of anticardiolipin antibodies [4,16].

The optimal treatment is still not determined. Common recommendations consist of bed rest, analgesia, and adequate glycemic control. According to Onyenemezu and Capitle’s article [13], supportive care modalities can have an impact on full recovery time and the potential risk of DMI recurrence. The authors focus, in their analysis, on three basic techniques: surgery, physiotherapy, and bed rest. The first one, surgery, statistically seems to involve a noticeably higher average time to symptom resolution, mostly three times compared to bed rest, and an increased rate of recurrence—over 50% of studied cases after 30 days of the initial episode. Although, the excision of the lesion sometimes results in immediate pain relief, it should not be recommended. The role of physiotherapy remains uncertain. It was observed that physiotherapy in the acute phase of DMI can bring about an elongation of time of recovery (almost two times in comparison to bed rest); however, in some cases, that correlation has been dismissed. Bed rest resulted in the shortest recovery time (48 days on average) and the longest duration of the good interval before potential recurrence, which happened in approximately 25% of studied cases [4,13].

Other therapeutic measures included nonsteroidal anti-inflammatory drugs, aspirin and pentoxyphyline, nifedipine, and dipyridamole—medications that could influence a hypothetical cause of DMI such as diabetic vasculitis or hypercoagulability, and lead to a vasodilatation effect [11,14].

The short-term prognosis of DMI is good. Patients recover within a few weeks to even 6 months, mostly without functional impairment. Nonetheless, the recurrence is frequent and the further prognosis, referring to a general condition of a patient with end-organ microvascular disturbances, is rather unfavorable. Long-term survival can be variable [6,11,17,18].

## 4. Conclusions

Diabetic muscle necrosis is a complication that worsens the quality of life. A diagnosis requires additional specialistic procedures such as radiological examinations or a biopsy followed by a histopathological examination. This complication is a mild condition but may be recurrent.

## Figures and Tables

**Figure 1 ijerph-20-03390-f001:**
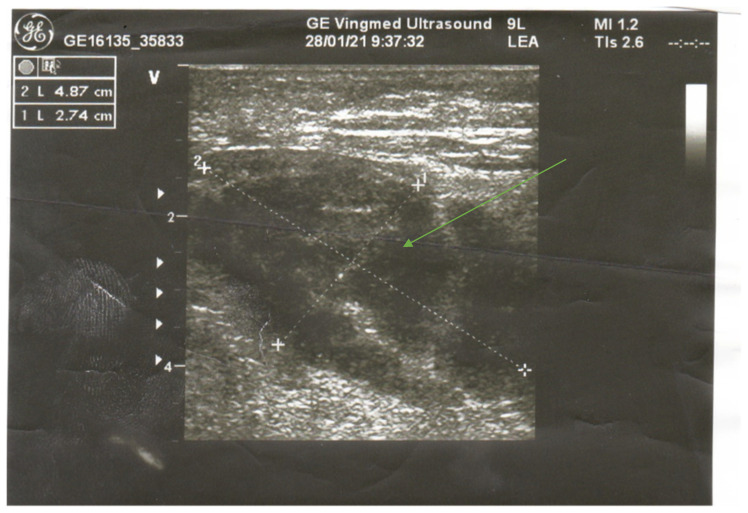
Oval lesion revealed in USG of the right thigh.

**Figure 2 ijerph-20-03390-f002:**
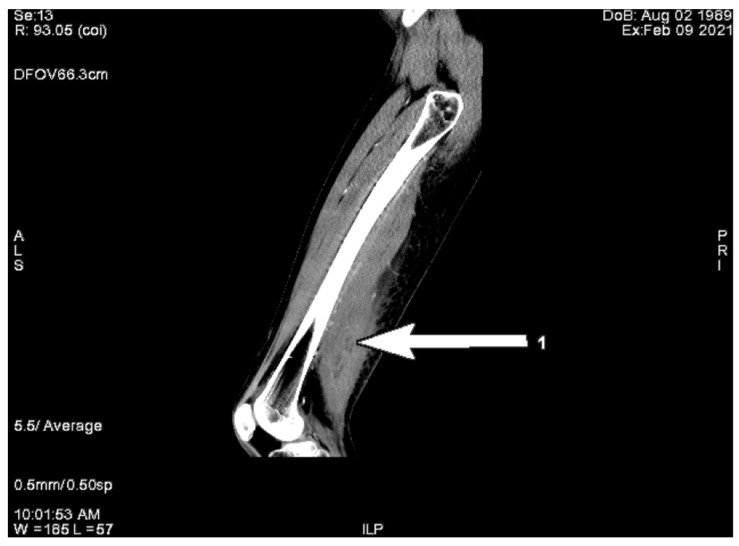
Right thigh CT after contrast administration revealed the fluid collection of degradation in posterior muscle compartment—arrow.

**Figure 3 ijerph-20-03390-f003:**
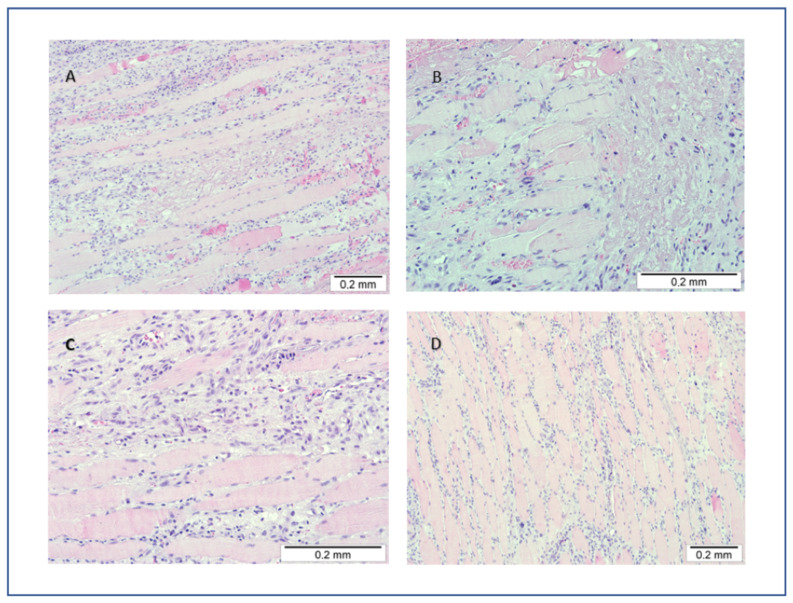
Histopathological presentation. (**A**) Areas of muscle necrosis and edema with extravasation of blood (HE, ×100). (**B**) Necrotic muscle fibers with focal replacement of necrotic fibers by loose fibrotic tissue (HE, ×200). (**C**) Early fibrotic tissue with mononuclear cell infiltration (HE, ×200). (**D**) Signs of muscle fiber regeneration with scanty lymphocytic infiltrates (HE, ×100).

**Table 1 ijerph-20-03390-t001:** The results of the patient’s laboratory test at the beginning of hospitalization and in the follow-up.

Laboratory Test (Units):	Reference Range:	On Admission:	Control (Acute Phase of DMI):
Hemoglobin (g/dL)	14–18	8.7	8.8
Hematocrit (%)	40–54	27	33.1
Platelets (10^3^/uL)	140–440	250	350
White blood cells (WBC) (10^3^/uL)	4–10	8.61	9.48
Erythrocyte sedimentation rate (mm/h)	1–10	51	-
High-sensitivity C-reactive protein (mg/L)	0–5	17.2	Acute phase: 136.5 At the end of hospitalization: 5.39
Hemoglobin A1c (%)	4–6	8.3	-
Total cholesterol (mg/dL)	130–200	123	-
HDL cholesterol (mg/dL)	min. 40	66	-
LDL cholesterol (mg/dL)	0–135	47	-
Triacylglycerol (mg/dL)	0–150	51	-
Creatinine (mg/dL)	0.8–1.3	6.7	4.2
Parathyroid hormone pg/mL	15–68.3	164.9	-
Calcium (mg/dL)	8.8–10.6	9.4	9.2
Phosphorus (mg/dL)	2.4–4.5	4.6	4.5
Creatine kinase (U/L)	0–171	-	94
Albumin (g/dL)	3.5–5.6	3.5	-
Alanine transaminase (U/L)	0–45	11	8
Aspartate transaminase (U/L)	0–35	14	14

## Data Availability

The data are available upon request.

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
