# Peer review of "Diabetic Muscle Infarction—A Rare Diabetic Complication: Literature Review and Case Report"

_ijerph, 2023, doi:10.3390/ijerph20043390_

Round 1

Reviewer 1 Report

Please improve matherial and method section.

Author Response

Thank you very much for your review. We have taken all reviews into consideration. We have updated and detailed the patient description and laboratory results in table 1.

According to the other reviews, we have removedmatherial and method section and created more specific status praesens.

Reviewer 2 Report

Against the background of some literature review, the authors describe a case of „diabetic muscle infarcton“ in a 31 year old male patient with long term type 1 diabetes of 25 years requiring dialysis because of end-stage renal disease.

Though not uninteresting as a case, the characterization of the patient phenotype needs major optimization and detailed granularity:

1)      Was the patient bedridden and unnoticed injury (because of neuropathy) may have occurred in the short head of the m. biceps fem. while encountering serious difficulties during forced change of position in bed due to muscular weakness?

2)      What about pressure injury?

3)      What about the neurological status, especially of the lower limb, including assessment of muscle strength? Obviously, the patient suffered from autonomic neuropathy (gastroparesis).

4)      What about description and classification of diabetic retinopathy as a surrogate of overall microvascular damage? Obviously, the patient also had widespread medial sclerosis of the arterial system.

5)      Use of statins or other drugs which may impact muscular integrity or confer prior muscular damage?

6)      BMI, cachexia, frailty measures?

7)      Serum concentrations of Ca, phosphate, parathyroid hormone? Secondary hyperpara?

8)      Serum albumin, hematocrit, platelet count?

9)      ECG, cardiac rhythm, blood pressure?

Author Response

Thank you very much for your review. We have taken all reviews into consideration. We have updated and detailed the patient description and laboratory results in table 1.

“We present a case of a 31-years old patient with type 1 diabetes diagnosed at the age of 6. Diabetes is complicated with neuropathy - a symmetrical distal sensory polyneuropathy with loss of sensation of touch, vibration, temperature and pain was diagnosed three years ago. Its complication is Charcot osteoarthropathy, for which reason the patient is advised to relieve pressure on his lower limb and uses a wheelchair outside the home. At home, the patient moves around independently, but due to his advanced retinopathy, he often suffers small bruises, especially on his lower extremities.

Patient was also diagnosed with proliferative diabetic retinopathy (PDR) in both eyes. Due to the rapid progression of this microvascular complication the patient lost his vision in the left eye.  His poor visual acuity was also considered to be the reason of mobility problems, increased number of injuries and difficulties with applying dressings on the diabetic foot ulcers. End-stage renal disease as a complication of nephropathy was diagnosed. Patient is on hemodialysis continued for 3 years

 Currently, the patient has been admitted to the diabetes ward due to inadequate diabetes control. He has developed postprandial hyperglycemia but also hypoglycemia that occurred after meals and varied from 1,3 – 26,7 mmol/L. His

glycosylated hemoglobin was 8.3% which was not reliable because of anemia and recurrent symptomatic sever  hypoglycemic episodes provoked by even low doses of insulin. Patient reported frequent use of glucagon to avoid hypoglycemia.

On admission to hospital, the patient was in a good general condition, with good cardiovascular and respiratory capacity, and on physical examination there was a deformity of the left foot with a small (20 x 10 mm) ulceration present on the sole of the foot in the midfoot area, with no signs of infection - a picture of Charcot osteoarthropathy. In addition, a slight swelling of both lower extremities was found, which we associate with dialysis therapy and a sedentary position. Small scratches were present on the skin of the lower extremities from small injuries sustained at home. There were no pressure sores or other skin ulcers. Muscle strength of the upper and lower limbs was normal and symmetrical. Pulse present in typical areas

Heart rate was steady on ECG normal recording. Above the lungs without stasis, abdomen soft, non-painful, at chest level. Calculated BMI 25.0. Blood pressure was not well controlled, measured on admission was 200/98 mmHg. His laboratory results are presented in table 1.”

Reviewer 3 Report

Dear authors, 

Indeed, the case is worthmentionted. Kindly have some comments below:

Abstract: Kindly ask you to update the abstract. More specific, the abstrasct should include also the case of the patient.

Introduction: I strongly suggest to update the introduction, adding the causes of muscle infarction, beyond Diabetes.

Materials and Methods section is not included in case reports.

Figure 1: please add arrow

CT, MRI, ESRD, STIR: please explain the abbreviation, the first time you mention it in text

«We replaced previous treatment by introducing small doses of insulin injected more frequently with singinficant improvement»: you do not mention the previous treatment, or the new treatment.

“Patient denied any trauma. “: the message is not clear

“After biopsy, the patient reported significant pain relief and as result analgesics were discontinued.”: the way the sentence is written, seems that biopsy is the cure! Kindly ask you to modify.

General comment: Why we diagnosed Diabetic muscle infarction and not just muscle infarction? We other causes excluded?

Author Response

Thank you very much for your review. We have taken all reviews into consideration. We have updated and detailed the patient description and laboratory results in table 1.

We have also updated in the introduction other possible causes of DMI.

Section material and methods has been deleted.

Other things have been corrected as suggested, the sentence about feeling relieved after the biopsy has been removed.

Reviewer 4 Report

RabczyÅ„ski et al.  have presented a case report of Diabetic muscle infarction - a rare diabetic complication. Overall, the symptom, diagnosis and treatment of this case are informative. The literature review is adequate, and manuscript is well written. However, following issues should be addressed.

1.      The USG, CT and Histopathological presentation showed a “myonecrosis”. Although it was most likely due to the diabetic complication based on the patient’s history of long-term type I diabetes and poor glycemic control, there were no specific markers or direct pathological evidence in this case indicating a definitive link. Authors need to discuss it.

2.      Has Diabetic muscle infarction been reported in other muscle types such as smooth muscles or cardiac muscles?

3.      Did the histopathological results indicate that extravasation of blood in lesion was caused by hemorrhagic infarction or microvascular chronic inflammation?

Author Response

Thank you very much for your review. We have taken all reviews into consideration. We have updated and detailed the patient description and laboratory results in table 1.

  1. a) After USG doppler we do not confirmed vascular disorders like deep vein thrombosis, leg ischemia, post traumatic hemorrhage or aneurysms (patient denied prior trauma). Patient was not treated with statins. According to normal value of WBC, lack of fever, resting pain, local signs of infection like skin warmness and redness bacterial infection was not proved. treatment antibiotics were not effective.

Sudden severe symptoms onset was not typical for neoplasia. To exclude other inflammatory problems like myositis or rhabdomyonecrosis, amyotrophy we decide to take histopathological biopsy, which demonstrated diffuse muscle necrosis, edema, mononuclear cell infiltration and local muscle fiber regeneration.

  1. b) In our review of the literature, we found no information on spontaneous smooth muscle or myocardial necrosis in diabetes mellitus. There are reports that scelatal muscle infarction in a diabetic population has similar prognosis as compared with myocardial infarction. DMI is a very late stage of terminal diabetic complication.

  1. c) The presence of focal blood extravasation does not differentiate hemorrhagic infarction from microvascular inflammation. It could be ischemia reperfusion injury.

Round 2

Reviewer 2 Report

Thanks for an improved clinical description of the patient phenotype, though an exact point by point response was not provided.

Yes, the patient indeed is dependent on partial wheel-chair use and did experience multiple minor traumata at home due to very poor vision which warrants discussion whether an unrecognized trauma (due to advanced sensory neuropathy - the patient actually also suffers from a Charcot foot which is associated with a similar mechanism of unrecognized traumata) that my induce some intramuscular hemorrhage within the m. biceps femoris - a typical predilection spot for a wheel-chair related trauma! 

As I understand, the table now states an LDL-cholesterol (please correct to appropriate terminology) of 47mg/dl. Is the patient so extremely malnourished (what about serum albumin concentration, as asked before?) that his LDL-chol is that low without using a statin or any other lipid lowering drugs, as stated now twice in the text? Discussion needed.

Patient does have secondary hyperpara which should be stated in the text.

there are still multiple typos such as sever instead of severe or scelatal instead of skeletal etc. Please correct.

Author Response

Thank you very much for your reviews.
The response included in the word file.

Thank you for further suggestions to improve the quality of our publication.

Thanks for an improved clinical description of the patient phenotype, though an exact point by point response was not provided.

I'm sorry we didn't answer point by point before.

We have made corrections in the text.

Now we include the answer  below

Yes, the patient indeed is dependent on partial wheel-chair use and did experience multiple minor traumata at home due to very poor vision which warrants discussion whether an unrecognized trauma (due to advanced sensory neuropathy - the patient actually also suffers from a Charcot foot which is associated with a similar mechanism of unrecognized traumata) that my induce some intramuscular hemorrhage within the m. biceps femoris - a typical predilection spot for a wheel-chair related trauma!

It was very likely that the patient had suffered an  unnoticed  injury in this region. In the differential diagnosis, we considered abscess, neoplasm, myositis or hematoma. Hence, we conducted extensive imaging diagnostics. The performed procedures did not give a clear diagnosis, therefore a biopsy of the lesion was performed. The diagnosis of DMI was made after biopsy and histopatological evaluation of samples.

As I understand, the table now states an LDL-cholesterol (please correct to appropriate terminology) of 47mg/dl. Is the patient so extremely malnourished (what about serum albumin concentration, as asked before?) that his LDL-chol is that low without using a statin or any other lipid lowering drugs, as stated now twice in the text? Discussion needed.

Serum albumin concentration was 3.5 g/dl (n 3.5- 5.5 g/dl) at that time (assessed during hemodialysis procedure. Low LDL cholesterol concentration was observed for the long time without any statins. Patient is observed as an outpatient now and LDL level it is still low. His BMI is normal, there was no signs of cachexia. Low LDL -cholesterol and albumin concentration could be provoked by malnutrition caused by poor nutrition, which was possible due to the poor economic condition of the patient or from disorders that affect ability to absorb nutrients. In our patient, we identified the following conditions that can lead to malabsorption; gastroparesis and anemia.

In our opinion the above information are not included in the case report, because it seemed irrelevant to the description of the difficulties we encountered in the process of reaching the correct diagnosis.

We corrected terminology.

Patient does have secondary hyperpara which should be stated in the text.

The patient is on chronic dialysis and diagnosed with secondary hyperparathyroidism

There are still multiple typos such as sever instead of severe or scelatal instead of skeletal etc. Please correct.

We will correct typos in the course of further editing

1)Was the patient bedridden and unnoticed injury (because of neuropathy) may have occurred in the short head of the m. biceps fem. while encountering serious difficulties during forced change of position in bed due to muscular weakness?

2)What about pressure injury?

It was very likely that the patient had suffered an  unnoticed  injury in this region. In the differential diagnosis, we considered abscess, neoplasm, myositis or hematoma. Hence, we conducted extensive imaging diagnostics. The performed procedures did not give a clear diagnosis, therefore a biopsy of the lesion was performed. The diagnosis of DMI was made after biopsy and evaluation of samples.

3)What about the neurological status, especially of the lower limb, including

assessment of muscle strength? Obviously, the patient suffered from autonomic

neuropathy (gastroparesis).

Diabetes is complicated with neuropathy - a symmetrical distal sensory polyneuropathy with loss of sensation of touch, vibration, temperature and pain was diagnosed three years ago. Its complication is Charcot osteoarthropathy, for which reason the patient is advised to relieve pressure on his lower limb and uses a wheelchair outside the home.

4)What about description and classification of diabetic retinopathy as a surrogate of overall microvascular damage? Obviously, the patient also had widespread medial sclerosis of the arterial system.

Patient was also diagnosed with proliferative diabetic retinopathy (PDR) in both eyes. Due to the rapid progression of this microvascular complication the patient lost his vision in the left eye. His poor visual acuity was also considered to be the reason of mobility problems, increased number of injuries and difficulties with applying dressings on the diabetic foot ulcers.

5)Use of statins or other drugs which may impact muscular integrity or confer prior muscular damage?

Statin were not used, low LDL cholesterol level were observed

6)BMI, cachexia, frailty measures?

BMI 25, no signs of cachexia, frailty measures were not assessed

7)Serum concentrations of Ca, phosphate, parathyroid hormone? Secondary

hyperpara?

Ca  9,4  mg/dl, phosphate   4,6 mg/dl, PTH 164,9 pg/mL

The patient is on chronic dialysis and diagnosed with secondary hypoparathyroidism

8)Serum albumin, hematocrit, platelet count?

Albumin 3,5 g/dl hematocrit 27% platelet count 250(10*3/uL)

9)ECG, cardiac rhythm, blood pressure?

Heart rate was steady on ECG normal recording

Blood pressure was not well controlled, measured on admission was 180/100 mmHg.

thank you again for your reviews

Reviewer 3 Report

Kindly ask you for a point to point answer.

Author Response

Thank you very much for your reviews.
The response included in the word file.

  1. Abstract: Kindly ask you to update the abstract. More specific, the abstrasct should include also the case of the patient

The abstract has been updated and made more personalised.

  1. Introduction: I strongly suggest to update the introduction, adding the causes of muscle infarction, beyond Diabetes.

“The pathogenesis of diabetic muscle infarction is not fully known. Possible causes include, in addition to poorly compensated diabetes and its complication of vascular microangiopathy, arteriosclerosis with embolic material from ulcerated plaques. Silberstein et al. proposed a mechanism based on hypoxia and reperfusion.”

  1. Materials and Methods section is not included in case reports.

As suggested, we have removed the material and methods section.

  1. Figure 1: please add arrow

Arrow has been added.

  1. CT, MRI, ESRD, STIR: please explain the abbreviation, the first time you mention it in text

Abbreviations were developed at the first possible moment.

  1. «We replaced previous treatment by introducing small doses of insulin injected more frequently with singinficant improvement»: you do not mention the previous treatment, or the new treatment.”

Now it’s:

“In the treatment of diabetes, the patient received insulin in an intensive insulin therapy regimen with basal insulin administered once a day and injections of prandial insulins. We introduced a division of prandial insulin, i.e. administration of half the dose before a meal and a control 2 hours after a meal with correction with significant improvement.”

 “Patient denied any trauma. “: the message is not clear

The sentence was inappropriate and has been removed. In another part of the text we refer to the possibility of trauma as a cause of DMI

“During next days of hospitalization, the patient reported sudden pain localized on the lateral, distal part of his right thigh. The pain occured at rest and was aggravated by movement. The patient denied any prior trauma such as a blow, pressure, stretch of this leg region.”

 “After biopsy, the patient reported significant pain relief and as result analgesics were discontinued.”: the way the sentence is written, seems that biopsy is the cure! Kindly ask you to modify.

Biopsy is not a treatment, the sentence has been deleted.

Thank you again for your reviews.

Round 3

Reviewer 2 Report

I did't see some of the discussion points raised in the previous review being addressed in the submitted revised ms

Author Response

We have included the remaining comments from the review in the manuscript. thank you very much for your review and help.

  1. End-stage renal disease was diagnosed as a complication of nephropathy. The patient has been on haemodialysis for 3 years and during this time secondary hyperparathyroidism was diagnosed.

  1. Serum albumin concentration was 3.5 g/dl. In addition, low LDL cholesterol had been observed for a long time without the use of statins. BMI was normal and no features of cachexia were observed. Low LDL -cholesterol and albumin concentrations may have been provoked by malnutrition, which may have been influenced by the patient's poor economic status, as well as disorders affecting the ability to absorb nutrients. In our patient, we identified the following conditions that could lead to malabsorption: gastroparesis and anaemia.

  1. It was very likely that the patient had suffered an unnoticed injury in this area. Hence, an extensive imaging diagnosis was carried out (USG, CT scan) and the procedures performed did not give a clear diagnosis. To exclude other inflammatory problems like myositis or rhabdomyonecrosis, and amyotrophy we decide to take a histopathological biopsy, which demonstrated diffuse muscle necrosis, edema, mononuclear cell infiltration and local muscle fiber regeneration. The diagnosis of DMI was made after biopsy and histopatological evaluation of samples.

Reviewer 3 Report

.

Author Response

We have updated the manuscript according to the reviewers' comments. thank you very much for your help and feedback.